# Investigating the Nature of 3D Generalization in Deep Neural Networks

## Abstract

Visual object recognition systems need to generalize from a set of 2D training views to novel views. The question of how the human visual system can generalize to novel views has been studied and modeled in psychology, computer vision, and neuroscience. Modern deep learning architectures for object recognition generalize well to novel views, but the mechanisms are not well understood. In this paper, we characterize the ability of common deep learning architectures to generalize to novel views. We formulate this as a supervised classification task where labels correspond to unique 3D objects and examples correspond to 2D views of the objects at different 3D orientations. We consider three common models of generalization to novel views: (i) full 3D generalization, (ii) pure 2D matching, and (iii) matching based on a linear combination of views. We find that deep models generalize well to novel views, but they do so in a way that differs from all these existing models. Extrapolation to views beyond the range covered by views in the training set is limited, and extrapolation to novel rotation axes is even more limited, implying that the networks do not infer full 3D structure, nor use linear interpolation. Yet, generalization is far superior to pure 2D matching. These findings help with designing datasets with 2D views required to achieve 3D generalization.

## 1 Introduction

Visual object recognition systems need to be able to generalize to novel views of objects using a small set of training views. This ability has been studied extensively in neuroscience, psychology, and computer vision (Poggio & Edelman, 1990; Logothetis et al., 1995; Riesenhuber & Poggio, 1998; Cooper et al., 1992; Biederman, 2000).

In order to explain these remarkable generalization capabilities of the human visual system in terms of object recognition, a range of different hypotheses have been proposed. One common theory is that humans perform pure view-based recognition (Poggio & Edelman, 1990; Logothetis et al., 1995; Riesenhuber & Poggio, 1998). However, the generalization capabilities exhibited by humans in visual recognition are hard to explain using a purely view-based recognition system. An alternative view is that humans build a partial 3D representation of the object rather than leveraging purely view-based recognition (Cooper et al., 1992; Biederman, 2000; Biederman et al., 1993).

Deep learning models have also demonstrated the capability to recognize objects from novel viewpoints. However, since these mechanisms are not explicitly built into the architecture, the reasons responsible for this capability are unclear. This knowledge is important to understanding and improving visual recognition systems.

Based on prior work in neuroscience and computer vision, we identify three distinct classes of behaviors for visual recognition that can explain the generalization capabilities of deep learning models:

- Full 3D recognition, where the system recovers a full 3D model of the object given a limited number of views. Once this 3D object is constructed, recognition is performed by selecting the model which

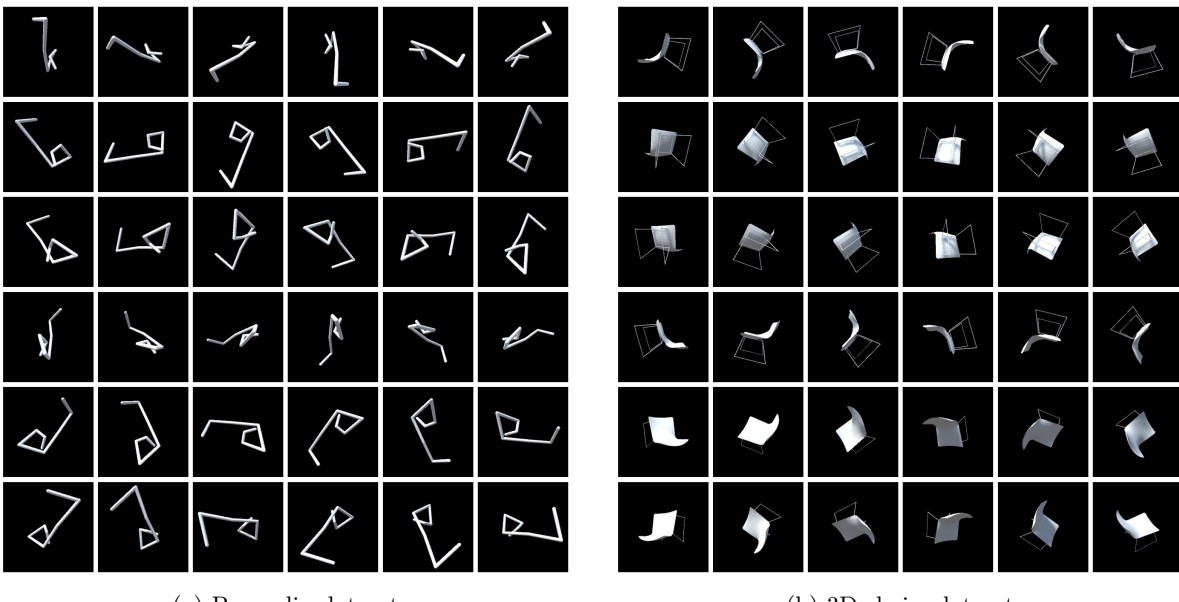

(a) Paperclip dataset          (b) 3D chairs dataset

Figure 1: **Examples from the generated datasets.** (a) **Paperclip dataset** where we plot a single paperclip object at different orientations along the z-axis on the horizontal axis and along the y-axis on the vertical axis. See Fig. 12 for rotation along other axes. (b) **3D chairs dataset** where we use the same generation protocol as for the paperclip dataset but use the 3D models of chairs from ShapeNet (Chang et al., 2015) instead of synthetic Paperclips.

best explains the given view. Such a model should generalize to all views given a small number of training views.

- Pure 2D matching, where recognition is performed directly by identifying the nearest training view for the given test view. Such a model should generalize only to similar views, independent of the axis of rotation.

- Matching based on a linear combination of views (2.5D matching), where the system interpolates between a limited number of views in order to construct a better representation of the underlying 3D object. Such a model should generalize well to rotation axes represented in the training data.

In this paper, we aim to understand the mechanisms responsible for generalization in deep networks by evaluating how far a model can generalize by learning on a limited number of training views for a given 3D object. 2D views of objects are generated via rotations around different axes.

The appearance of 3D objects varies across viewpoints for two reasons: self-occlusion, and the geometry of 3D projection. Self-occlusion results simply in the presence, or absence of features in the image, similar to other forms of occlusions. The geometric dependence of the two-dimensional view on the pose parameters has received extensive study in the literature (Plantinga & Dyer, 1990; Cyr & Kimia, 2001; Gigus & Malik, 1990). Synthetic wire-like objects i.e., Paperclips are a particularly good model to study the latter phenomenon as they have limited self-occlusion. For this reason, Paperclips have been used to study generalization in humans, monkeys, and pigeons in the past (Poggio & Edelman, 1990; Logothetis et al., 1995; Riesenhuber & Poggio, 1998; Spetch et al., 2006).

We use a limited number of views to train and evaluate the generalization capabilities of the model on all possible axis-aligned rotations. We also perform evaluations on 3D models of real-world objects – models of chairs from ShapeNet (Chang et al., 2015).

We formulate this as a supervised classification task, where the 3D model refers to the class, while specific 2D views based on different 3D rotations of the underlying 3D object refer to the training/evaluation examples.

This formulation is consistent with experiments on monkeys/humans where they were tasked to distinguish views of the same object from views of another object (Poggio & Edelman, 1990; Logothetis et al., 1995; Riesenhuber & Poggio, 1998; Spetch et al., 2006).

Our results show that deep models exhibit behavior that is neither full 3D recognition nor pure 2D matching. This behavior is also distinct from the linear combination of views on paperclips in subtle ways. In particular, the contributions of this paper are:

- We present new datasets of synthetic paperclips and chairs for testing the 3D generalization of models.

- By analyzing the generalization capabilities of deep learning models on rotations of 3D objects around different axes, we show that deep learning models are capable of substantial generalization across views, but behave differently from all existing mathematical models of generalization across views.

- We show that generalization improves with the number of classes, showing that 3D generalization is based in part on learning model-independent features.

- We show that these results are consistent across different input representations, architectures (ResNets, VGG, and ViTs) as well as real-world 3D objects (3D models of chairs (Chang et al., 2015)), highlighting that these results are not an artifact of the chosen representation, 3D object, or architecture.

## 2 Background & Related Work

We analyze the 3D generalization capabilities of deep learning models, as well as the class of behavior leveraged by these models for recognition by training on different 2D views of the underlying 3D object. For this purpose, we first describe the geometry of vision, followed by different classes of behavior for visual recognition that we consider in this work to evaluate deep networks. We then describe visual recognition in humans, highlighting the behavioral class that they are hypothesized to follow. Finally, we discuss prior work analyzing failure modes of deep networks in 3D generalization as well as the evaluation of generalization on OOD poses which is closest to our work.

### 2.1 Geometry of Vision

Visual systems are commonly modeled as performing a central projection of 3D objects onto the image plane. This projection is determined by 6 pose parameters covering 3D translation and rotation, where pitch, yaw, and roll correspond to rotation along the x-axis, y-axis, and z-axis respectively. For objects sufficiently distant from the camera, this projection is usually approximated via an orthographic projection along with the scale. Based on these 6 pose parameters, the set of views forms a 6-dimensional manifold in image space.

An ideal recognition system should be invariant to variations in all 6 parameters. We can achieve approximate invariance to 4 of the 6 parameters by modeling recognition to be invariant under 2D translation, rotation, and scale. This is commonly achieved in deep learning systems via data augmentation. This leaves 2 pose parameters, namely rotation along the x-axis and y-axis (pitch and yaw).

### 2.2 Classes of Behavior for Visual Recognition

We consider three dominant classes of behavior for visual recognition from classical computer vision as well as neuroscience literature i.e. (i) full 3D recognition, (ii) pure 2D recognition, and (iii) 2.5D matching based on a linear combination of views.

**Full 3D Recognition.** Huttenlocher & Ullman (1987) proposed a pure 3D recognition algorithm i.e. recognition by alignment where the 3D model is aligned with the given 2D view for recognition. This assumes access to an underlying 3D model of the object. For full 3D recognition, the manifold of views can

be reconstructed from a limited number of samples using structure from motion theorems (Jebara et al., 1999). Systems performing full 3D recognition should generalize to all views given just a couple of training views.

**Pure 2D Matching.** Poggio & Edelman (1990) presented pure 2D matching systems, where distances to views are defined via radial basis functions (RBF) from existing views, and recognition is performed based on the nearest training view for the given test view. For pure 2D matching, 4 of the 6 view parameters can be eliminated via augmentation, followed by simple nearest neighbor lookup or sampling for the remaining two parameters. Systems performing pure 2D matching should generalize to very similar views, independent of the axis of rotation.

**Linear Combination of Views.** Linear combination of views (or 2.5D) matching was popularized by Ullman & Basri (1989), who showed that given 2D image views from an underlying 3D model $\mathcal{M} = \{M_1, M_2, ..., M_n\}$, a given 2D image $P$ can be recognized as an instance of the object $\mathcal{M}$ if, for some constraints $\alpha_i$, we have $P = \sum_i \alpha_i M_i$. The model, in this case, is represented by the coordinates of the object's silhouette (an edge-map). They established that such a method can handle 3D transformations of an object, assuming that the object has been correctly segmented from the background. This approach is akin to interpolating the given views to achieve a larger effective number of views for recognition. In the context of the linear combination of views, the 6D view manifold can be approximated with a linear manifold that contains the original view manifold. Systems performing linear combinations of views should generalize well to rotations along the axes represented in the training data.

### 2.3 Visual Recognition in Humans

There has been a long line of research arguing for view-centric or pure 2D matching-based recognition in humans (Poggio & Edelman, 1990; Logothetis et al., 1995; Riesenhuber & Poggio, 1998), where the visual system responds to view-specific features of the object, which are not invariant to different transformations of the original object.

The generalization-based approach is another dominant hypothesis where the visual system learns view-invariant features which can be useful for recognition even when considering novel views of the object (Cooper et al., 1992; Biederman, 2000; Biederman et al., 1993). One hypothesis for how this can be achieved is by relying on Geon Structural Descriptions (GSD) of the object, which remains invariant to these identity-preserving transformations (Biederman et al., 1993).

Karimi-Rouzbahani et al. (2017) compared recognition between humans and deep learning systems, and argued that humans focus on view-invariant features of the input, while deep learning models employ a view-specific recognition approach, which impedes their ability to be invariant to a certain set of transformations.

### 2.4 3D Generalization Failures of Deep Networks

Recent work has also attempted to analyze the 3D generalization failures of deep networks. Alcorn et al. (2019) showed that novel 3D poses of familiar objects produced incorrect predictions from the model. Abbas & Deny (2022) extended this evaluation to state-of-the-art models and showed that current models still struggle to recognize objects in novel poses. Madan et al. (2021) similarly showed that such misclassification can be induced by even minor variations in lighting conditions or poses which can still be considered in-distribution. Experiments in psychology have also demonstrated that reaction times, as well as error rates, also increase for humans when looking at objects from novel viewpoints (Spetch et al., 2006).

The investigation of adversarial examples has also been extended beyond $L_p$ norm-based attacks using a differentiable renderer to identify adversarial poses (Liu et al., 2018; Zeng et al., 2019).

All these investigations are targeted toward trying to find vulnerabilities in the model recognition behavior. On the other hand, we attempt to understand the model's generalization when provided with a small number of training views.

### 2.5 Generalization to OOD poses

Schott et al. (2021) generated datasets based on a fixed generative model, and showed that the model fails to learn the correct latent factors of variation, hence, struggling to generalize in OOD scenarios regardless of the level of supervision used to train the model. However, their analysis is mainly limited to 2D generalization while we focus on 3D generalization.

Cooper et al. (2021) attempted to systematically understand the generalization of the model to novel poses by evaluating on out-of-distribution poses of the underlying 3D object. Although their work shares some conceptual similarities with ours, there are noteworthy differences in the evaluation settings:

- The study by Cooper et al. (2021) focused on OOD performance. In contrast, our research primarily addresses the issue of generalization from multiple training views (multi-view generalization).

- Our work is the first to demonstrate the dependence on the number of views, specifically the interaction between views. The observed synergistic effects of training on multiple views of the same object are quite intriguing.

- The conclusion drawn from Cooper et al. (2021) regarding the model's ability to achieve only planar generalization is related to our case of linear interpolation of views. However, our work extends this concept to multiple views.

- The experimental setup in Cooper et al. (2021) differs from ours, as they appear to perform per-category classification, rendering the classification problem to be considerably simpler. In our study, we formulate a per-object classification problem (similar to the prior studies in psychology and neuroscience (Poggio & Edelman, 1990; Logothetis et al., 1995; Riesenhuber & Poggio, 1998; Spetch et al., 2006)) and reveal the synergistic effects of the number of views as well as the number of classes on generalization across the training axis.

## 3 Methods

We first discuss the dataset generation technique employed in this paper for generating our datasets, followed by the training protocol used to train our models.

### 3.1 Datasets

The paperclip dataset is comprised of 10,000 synthetically generated 3D paperclip models. Vertices of the different 3D models are initialized randomly within a sphere around the previous point and connected together to form a paperclip. We restrict the number of vertices to 8 and reject 3D models with extremely sharp edges or overlaps between wires of the paperclip. Each paperclip is rescaled and translated to the center of the camera once the full paperclip has been generated. We then rotate the object for a full 360 degrees sequentially through all three axes. For multi-axis rotations, we consider a stride of 10 degrees, resulting in a total of $36 \times 36$ frames per combination. Since there are three such combinations in total (xy, xz, and yz), the total number of frames in the dataset turns out to be: $(360 \times 3 + 36 \times 36 \times 3) \times 10000 \sim 50M$. Examples from the dataset are visualized in Fig. 1a. Training is performed by sub-sampling these classes, as well as the number of views used during training.

We have also generated a similar dataset to compute 3D generalization on real-world 3D objects, specifically 3D models of chairs from ShapeNet (Chang et al., 2015). We render these objects without texture in order to ensure that the model is forced to use shape rather than texture for recognition. The chairs are randomly rotated to avoid them having in their canonical pose. This helps in breaking object symmetries, making them comparable to our paperclip setup. Examples from the chair dataset are visualized in Fig. 1b.

We use Unity3D for our 3D renderings (Unity Technologies, 2005). We place the camera at a small distance, looking directly at the object located in the center of the frame. See Fig. 11 for an illustration of the camera setup.

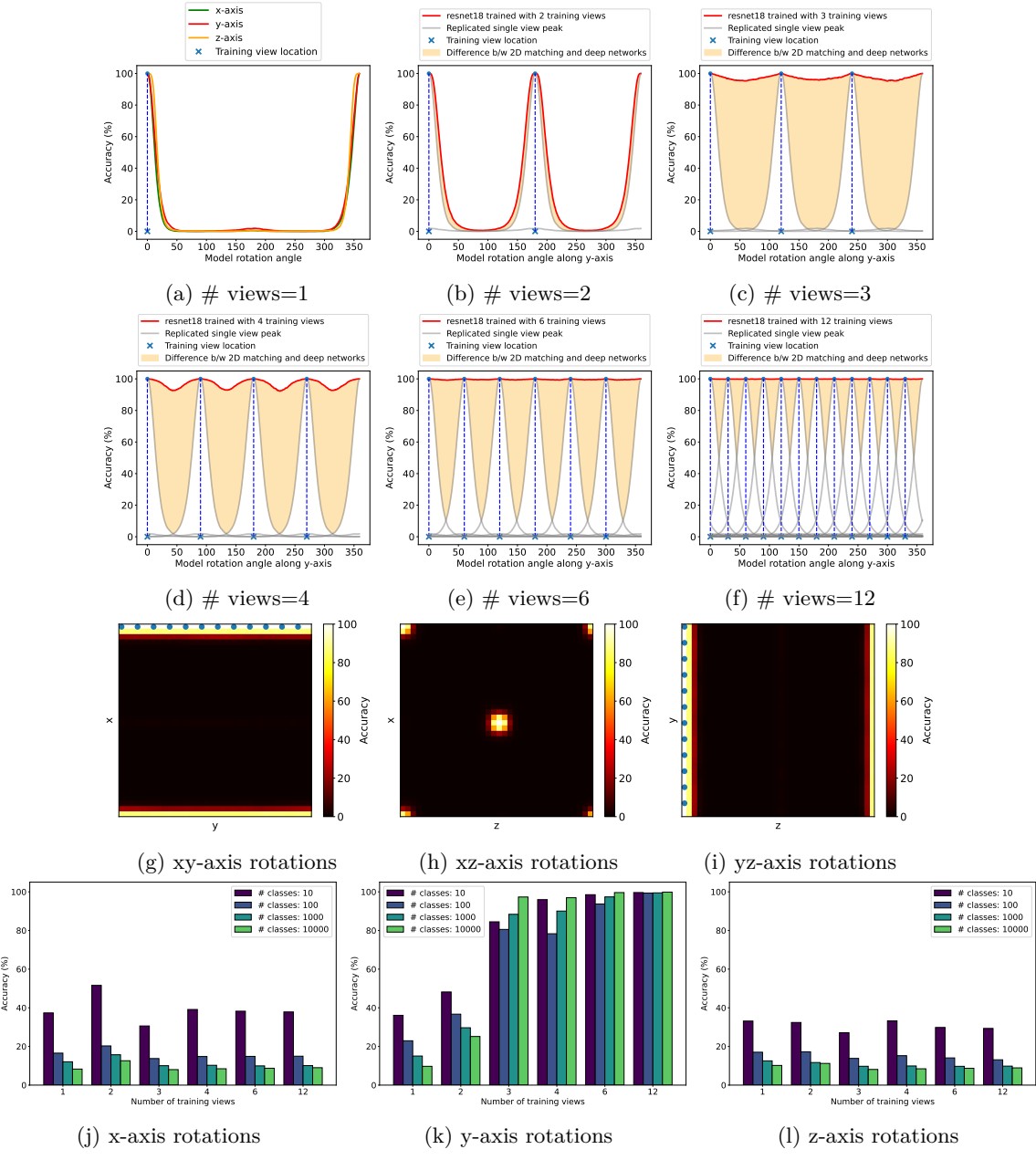

(a) # views=1    (b) # views=2    (c) # views=3

(d) # views=4    (e) # views=6    (f) # views=12

(g) xy-axis rotations    (h) xz-axis rotations    (i) yz-axis rotations

(j) x-axis rotations    (k) y-axis rotations    (l) z-axis rotations

Figure 2: **Change in recognition behavior with an increasing number of views.** We plot the performance of the model with an increasing number of views on our paperclip dataset when training with rotations only along the y-axis using 10000 classes, starting from a view with no rotation (0°). The red line indicates the observed performance, the gray line indicates the expected performance from a purely view-based model (replicated from # views=1 condition), and the orange region indicates the performance difference between a purely view-based recognition system and deep learning models. The second last row visualizes the performance of the most powerful setting (12 equidistant views w/ 10000 classes) when rotating along two different axes simultaneously with blue dots indicating training views. The final row summarizes these results with a different number of views as well as a different number of classes. The figure highlights how recognition behavior changes with an increasing number of views, where the model is strictly view-based when considering a single or two views (a-b) but exhibits surprising generalization to intermediate views (c-f) located in between when the number of views increases (≥ 3). The results also show that deep learning models do not generalize well to rotations around novel axes.

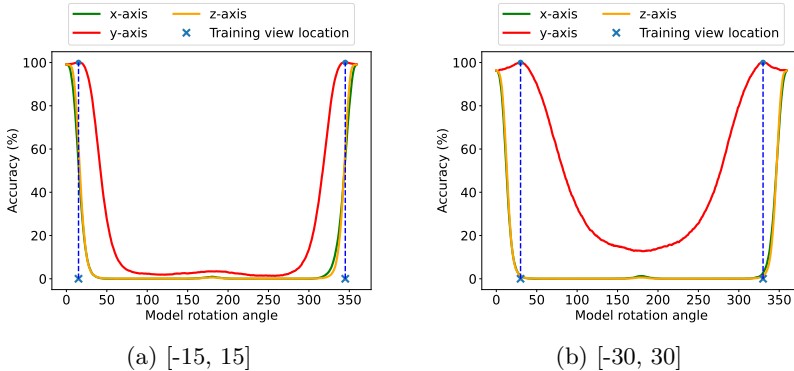

(a) [-15, 15]     (b) [-30, 30]

Figure 3: **Model achieves higher generalization on views embedded within training views.** We plot the performance of the model on two different settings i.e., with [-15, 15] as training views and [-30, 30] as training views. We see that despite 0-degree rotation and 30-degree rotation being equally spaced from training views in the case of [-15, 15], the model generalizes better to the views between training views i.e., 0-degree rotation.

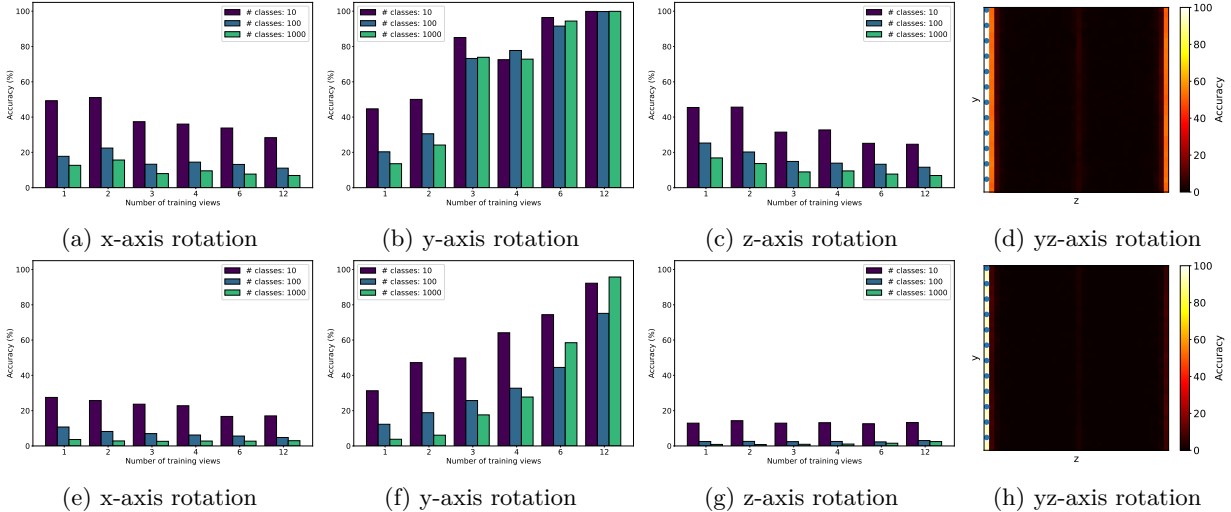

(a) x-axis rotation     (b) y-axis rotation     (c) z-axis rotation     (d) yz-axis rotation

(e) x-axis rotation     (f) y-axis rotation     (g) z-axis rotation     (h) yz-axis rotation

Figure 4: **Our results generalize to other representations.** We plot the performance of the model using different input representations with 1000 classes. The first row represents the coordinate image representation, while the second row represents the coordinate array representation (see Fig. 13 for a visual depiction of the representations used). The figure indicates that our findings generalize to other input representations.

## 3.2   Training

All training hyperparameters are specified in Appendix A and summarized in Table 1. We perform random horizontal flips and random cropping (with a scale of 0.5 to 1.0) during training following regular deep-learning training recipes. It is important to note that we only aim to capture the impact of 3D rotations. The model is expected to acquire translation and scale invariance via the random crops used during training (see Fig. 1a). This also avoids the learning of trivial solutions by the model.

We default to using ResNet-18 for our experiments unless mentioned otherwise. We always train on rotations along the y-axis and evaluate on rotations along all three axes, including simultaneous rotations along two axes.

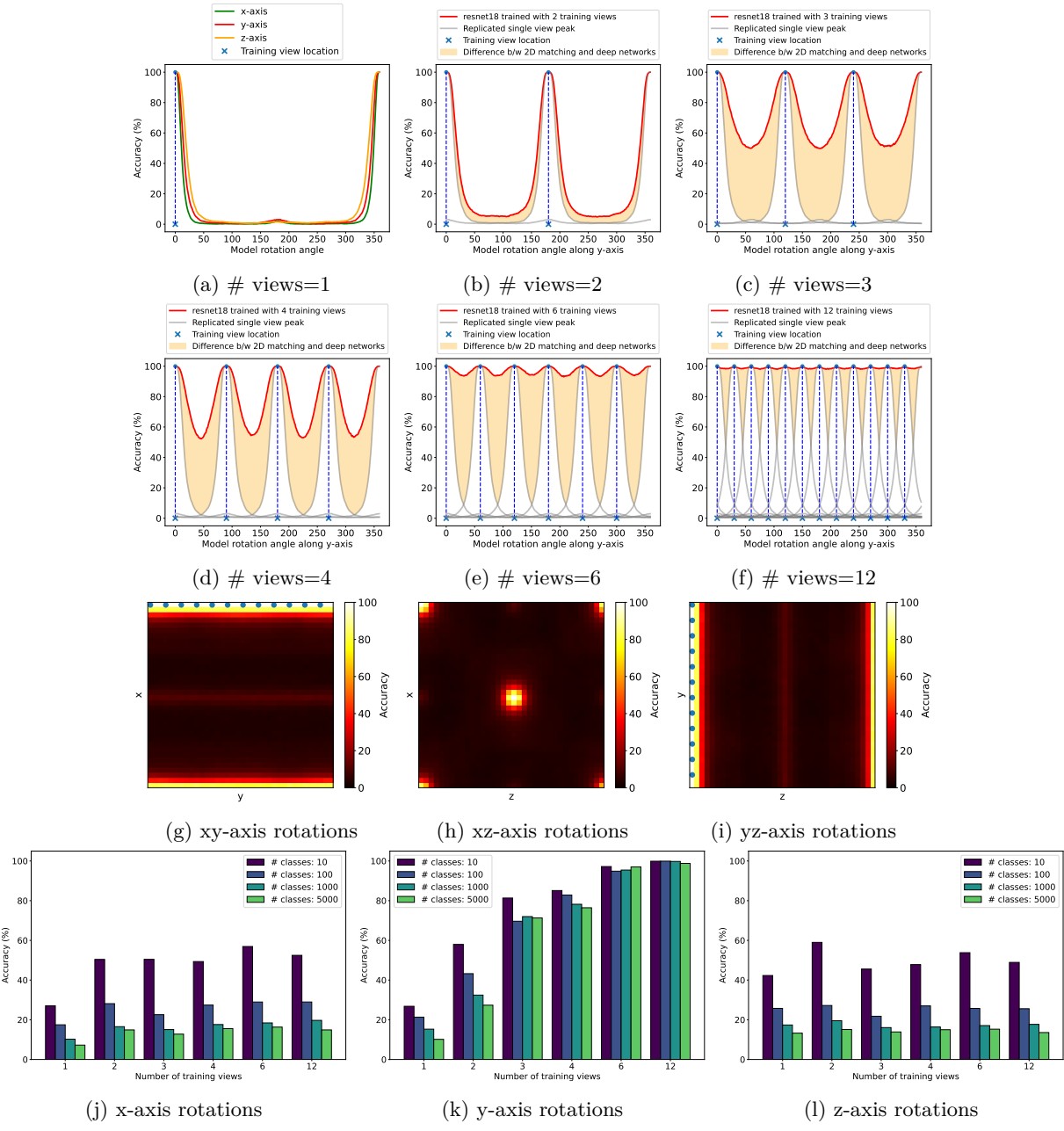

Figure 5: **Same conclusions hold for real-world 3D objects.** We plot the performance of the model on 3D chairs dataset (see Fig. 2 for description). We observe similar conclusions hold for 3D models where the model generalizes along the axis of rotation, but fails to generalize to rotations along novel axes.

# 4 Results

We divide the results into two distinct settings based on the range from which the views are sampled: (i) uniformly sampled views, and (ii) range-limited sampled views.

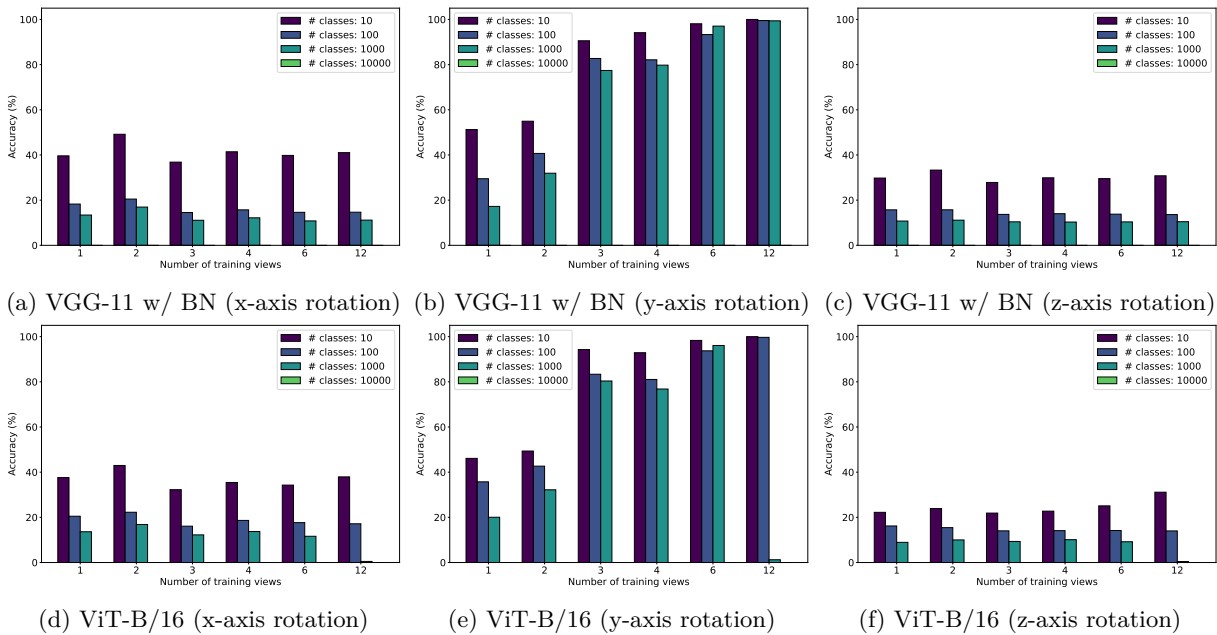

Figure 6: **Our results generalize to other architectures.** We plot the performance of the model with an increasing number of uniformly sampled views on the paperclips dataset for VGG and ViT (all previous results were based on ResNet-18). The figure indicates that our findings are not an architecture-specific artifact, but relate to the general behavior of deep learning models.

## 4.1 Uniformly sampled views

In this setting, we train the model on a limited number of uniformly sampled views from the view-sphere when only rotating along the y-axis. As the number of views increases, the model should get a holistic understanding of the full 3D object.

Fig. 2 presents our results in this setting, where we fix the number of classes to be 10,000. We highlight the difference in the performance of a purely view-based recognition system and our trained model with orange color. The model seems to be performing pure 2D matching with a small number of views (Fig. 2a-b). However, as the number of views increases, there is a drastic shift in recognition behavior where the model exhibits significantly better generalization than that of pure 2D matching (Fig. 2c-f).

*Is the model successfully recovering the full underlying 3D object?* In order to answer this, we evaluate the model's performance when rotating the object along a different axis. Full 3D recognition based on reconstructing a 3D representation of the object should generalize to novel rotations. However, we see in Fig. 2g-l that once we rotate the object along a different axis, the model fails to generalize.

These results disprove both full 3D recognition and pure 2D matching as possible classes of behavior employed by deep models, leaving only the linear combination of views as a possible class of behavior exhibited by deep models, which we will investigate further.

**Generalization on intermediate views between training views** Fig. 3 shows the performance of the model, helping us to understand generalization differences between views that are embedded within training views vs. views that are outside the set of training views. We see that despite the two views being equally distant from the training views, views that are embedded within the training views achieve higher generalization.

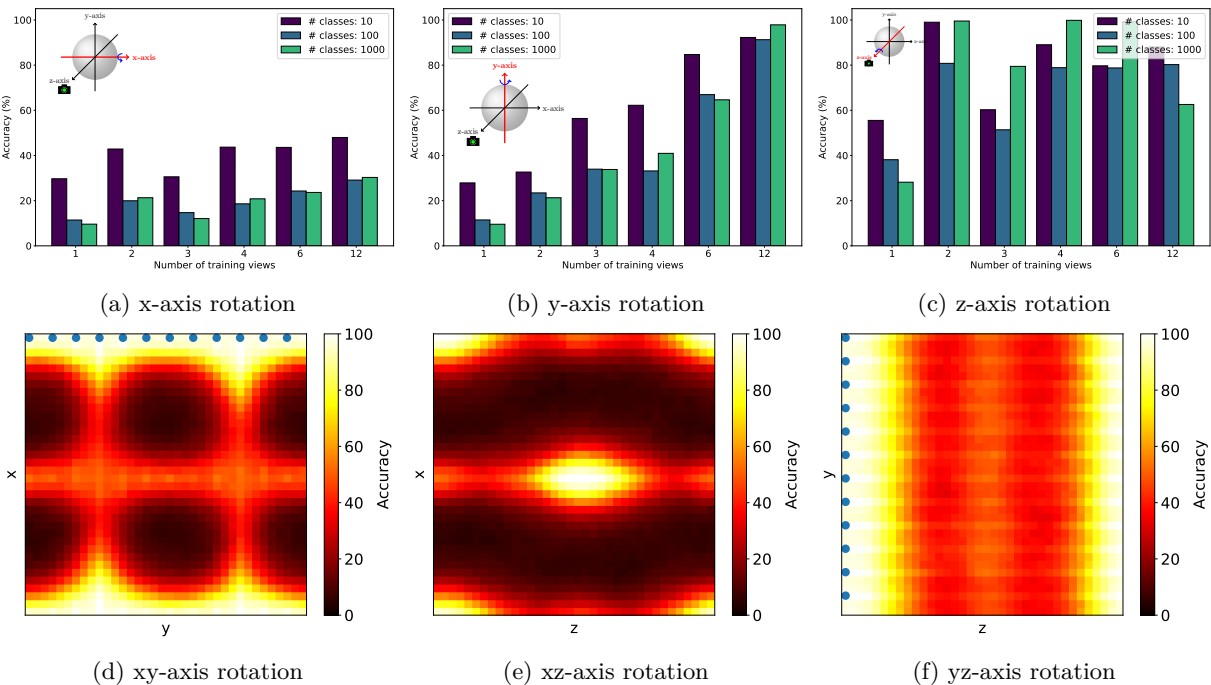

(a) x-axis rotation  (b) y-axis rotation  (c) z-axis rotation

(d) xy-axis rotation  (e) xz-axis rotation  (f) yz-axis rotation

Figure 7: **Random 2D rotation augmentation along the image plane improves generalization.** We plot the performance of the model along all three axes (see Fig. 11 for a visual illustration of the relationship between 3D rotations and the image plane). We see that random rotations in the image plane improve generalization along the z-axis which is partially aligned with the image plane but fails to improve generalization along a different axis i.e. x-axis. We also visualize multi-axis rotations in the second row for 12 training views and 1000 classes to get a better understanding of the model's generalization.

### 4.1.1 Generalization to other representations

Since we use 3D renderings of synthetic paperclip objects, this may make the process of recognition more difficult. A paperclip is fully specified by the position of its vertices. Therefore, we also evaluate two different simplified input representations i.e., (i) coordinate image representation and (ii) coordinate array representation. In the case of coordinate image representation, we represent the coordinates directly as a set of points on an image plane and train a regular CNN on top of these images. In the case of coordinate array representation, we project the coordinates onto two different arrays, one representing the x-coordinates and one representing the y-coordinates. As multiple points can be projected down to the same location, each vertex projection carries a weight of 1/8 for a total of 8 vertices. We concatenate the two arrays and train a multi-layer perceptron (MLP) on top of this representation. A visual depiction of these representations is provided in Fig. 13.

In particular, we train ResNet-18 on the coordinate image representation and a 4-layer MLP w/ ReLU activations on the coordinate array representation. This change in representation makes a negligible impact to the results as depicted in Fig. 4, where the model generalizes to intermediate views of the training axis, but fails to generalize to novel axes of rotation. These results indicate that our findings are not a representation-specific artifact.

### 4.1.2 Generalization to real-world objects

We visualize the results from our chairs dataset in Fig. 5. We observe a similar trend where the generalization of the model is higher than pure 2D matching but lower than full 3D generalization. These results advocate that the obtained results are not an artifact of the chosen object i.e. Paperclips, but rather, relate to the recognition behavior of deep learning models.

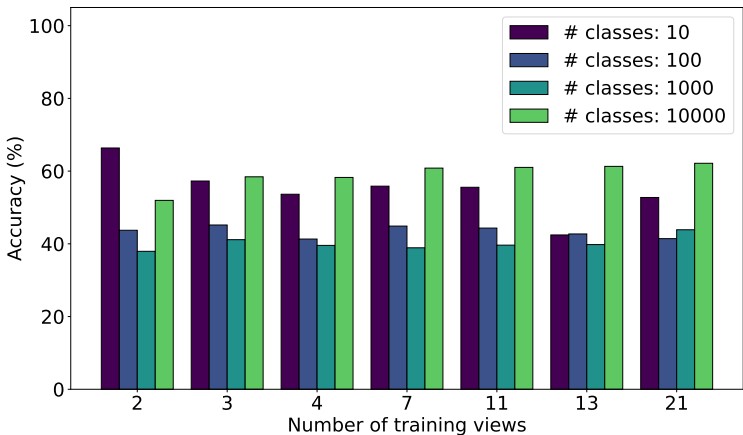

Figure 8: **Large number of views from a limited range fails to achieve generalization.** We plot the performance of the model with an increasing number of views sampled in the range [-30, 30] on the paperclips dataset when training and evaluating with rotations only along the y-axis using 10000 classes. The plot shows that the model fails to generalize to different rotations along the y-axis outside this limited range of views.

### 4.1.3 Generalization to other architectures

Fig. 2 presented generalization results for ResNet-18 (He et al., 2016). In Fig. 6, we additionally show results for VGG-11 (w/ BN) (Simonyan & Zisserman, 2014) and ViT-B/16 (Dosovitskiy et al., 2020). ViT-B/16 trained using 1000 classes failed to converge in our case, resulting in very low final accuracy. It is clear from the figure that our findings do generalize to architectures beyond ResNets. The plot also hints that the model is slightly better at generalizing towards rotation not directly in the image plane (x-axis) as compared to rotations directly along the image plane (z-axis). This highlights that in-plane rotations are particularly hard for deep-learning models to handle.

### 4.1.4 Generalization w/ 2D rotation augmentations

As the model particularly struggles to handle in-plane rotations, we evaluate the performance of the model by using in-plane rotations as an augmentation. In-plane rotation is equivalent to z-axis rotation as the camera is perfectly aligned with the rotation axis. The results are visualized in Fig. 7. We see that in-plane rotation improves generalization along the z-axis (which is aligned with in-plane rotations) while having no impact on generalization along the x-axis. This indicates that in-plane rotation augmentations alone are insufficient to force the model to learn a holistic 3D representation.

### 4.1.5 Robustness against changes in background

In order to ensure that the obtained results are not an artifact of the simple black background without any distractors, we replace the black background with a randomly chosen image from the landscapes dataset found on Kaggle[1] for every example in the batch as part of our augmentation pipeline (sample training images with the random backgrounds are visualized in Fig. 14). We find qualitatively similar results with these random backgrounds even though the task is considerably harder, resulting in higher error rates.

## 4.2 Range-limited sampled views

In this setting, we explore generalization from views sampled from a limited range of rotations. We train on a small subset of views sampled uniformly from [-30, 30] degrees of rotation and evaluate the performance of the model on all possible rotations. We visualize these results in Fig. 8. It is clear from the figure that

---

[1]https://www.kaggle.com/datasets/arnaud58/landscape-pictures

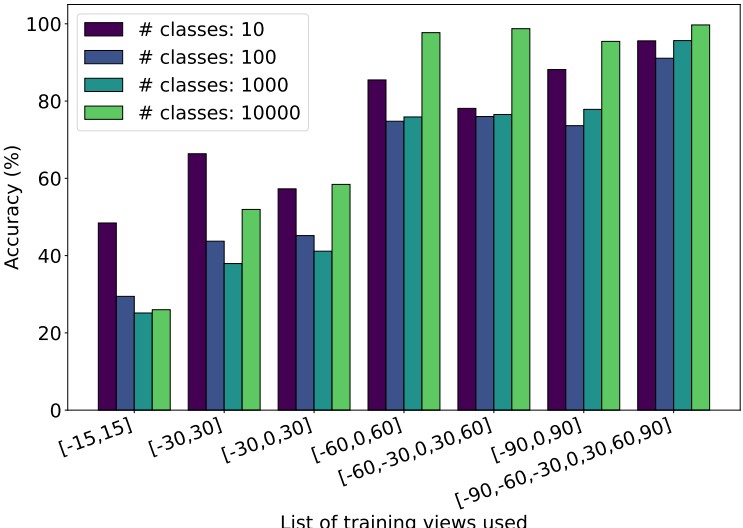

Figure 9: **Large view range is important for generalization.** We plot the performance of the model with an increasing range from which the views are sampled when training and evaluating with rotations only along the y-axis. The plot shows that the model ultimately achieves generalization to different rotations along the y-axis outside this limited range of views when increasing the range from which the views are sampled, specifically with a large number of classes. This hints that the model is leveraging knowledge across classes.

despite an increasing number of views sampled uniformly within this range, the model fails to generalize to views outside this range, indicating that the model failed to construct a view-independent representation of the input.

Results in Section 4.1 already ruled out the possibility of pure 2D matching or full 3D recognition, making the linear combination of views the most likely class of behavior employed by deep models for recognition. However, for objects such as paperclips, where self-occlusion is not an issue, a linear combination of views should be able to extrapolate i.e. generalize within the axis of rotation. Since this is not the case, this indicates that the model is operating somewhere close to the linear combination of views, but is still distinct in some aspects.

### 4.2.1 Extending the range of views

The model trained on range-limited sampled views fails to extrapolate to novel views along the training axis. In order to understand the relationship between view range and expected generalization along the training axis, we evaluate the performance of the model while relaxing the range limits used to train the model in Fig. 9. We see that once the range covers a large fraction of the possible views, and there is a sufficiently large number of views used for training, the model generalizes within the axis of rotation. However, this generalization is still limited to the training axis. Furthermore, this generalization is also correlated with the number of classes.

We further evaluate the relationship between the number of classes and generalization for a particular selection of training on views from [-60, 0, 60] degrees of rotation along the y-axis in Fig. 10. The plot shows that despite having the same number of views available for training the model, simply increasing the number of classes helps the model to generalize to rotations along the training axis by improving the model's performance, particularly on intermediate views between 100° to 260° rotation. This hints that the model is able to share knowledge between classes in order to construct a more general representation of the input.

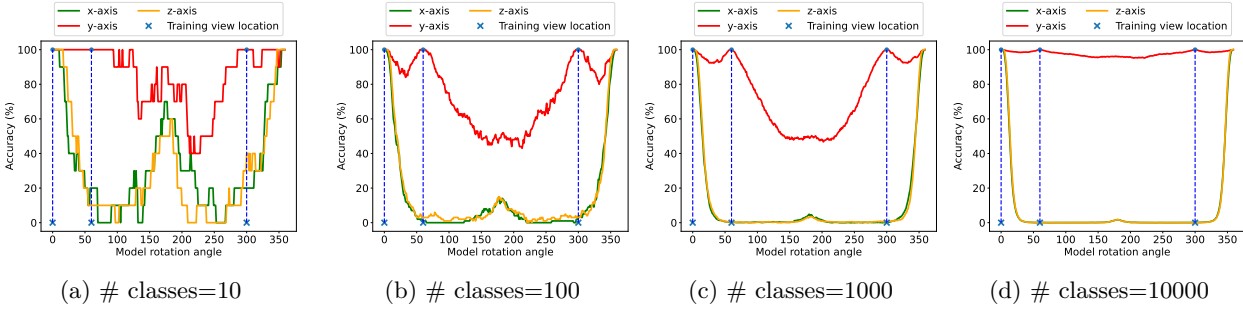

(a) # classes=10      (b) # classes=100      (c) # classes=1000      (d) # classes=10000

Figure 10: **Model benefits from a larger number of classes for generalization.** We plot the performance of the model with an increasing number of classes when training on only a particular set of rotations only along the y-axis per class ([-60, 0, 60]). Training views are marked with vertical lines. The plot shows that the model's generalization drastically improves with an increasing number of classes, specifically for intermediate views without any supporting training examples ($100° - 260°$), hinting that the model is leveraging knowledge across classes.

## 5 Conclusion

This paper has experimentally studied the behavior class employed by deep learning models for visual recognition by analyzing the generalization capabilities of the model on axis-aligned rotations of 3D objects when trained on a limited number of views.

Our results show that deep learning models generalize to novel views in a way that differs from all '*classical*' computer vision models. The closest model is matching based on a linear combination of views. We summarize our findings as:

- We have shown that deep models do not perform pure 2D matching since the achieved generalization is higher than a purely view-based recognition system

- We have shown that deep models do not perform full 3D recognition because the model failed to achieve generalization to rotations along novel axes

- We have shown that deep models behave similarly to linear interpolation of views, but are still distinct in terms of behavior as the model failed to generalize to rotations outside the limited range of views even for simple paperclip objects without self-occlusions

- We have shown that 3D generalization abilities generalize across different 3D models

These results are of practical importance in two ways. First, they allow us to design training sets for 3D object recognition with deep networks more efficiently by particularly focusing on views that are most beneficial for the model to learn. Our results show that covering entire axes is better than sampling more views from a limited range of rotations along the axes. Second, they imply that incorporating better 3D generalization capabilities directly into our deep networks such as object reconstruction followed by recognition may significantly improve the efficiency at which our networks use training data. This sample-efficient reconstruction can be achieved via structure from motion theorems (Jebara et al., 1999).

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

| Model | Epochs | Optimizer | Batch Size | Learning Rate | Momentum | Learning Rate Decay | Weight Decay | Gradient Clipping | Pretrained |
|-------|--------|-----------|------------|---------------|----------|---------------------|--------------|-------------------|------------|
| ResNet-18 | 300 | SGD | 128 | 0.1 | 0.9 | Cosine | 0.0001 | 10.0 | False |
| VGG-11 (w/ BN) | 300 | SGD | 128 | 0.01 | 0.9 | Cosine | 0.0001 | 10.0 | False |
| ViT-B/16 | 300 | SGD | 128 | 0.01 | 0.9 | Cosine | 0.0001 | 10.0 | True |

Table 1: Training Hyperparameters

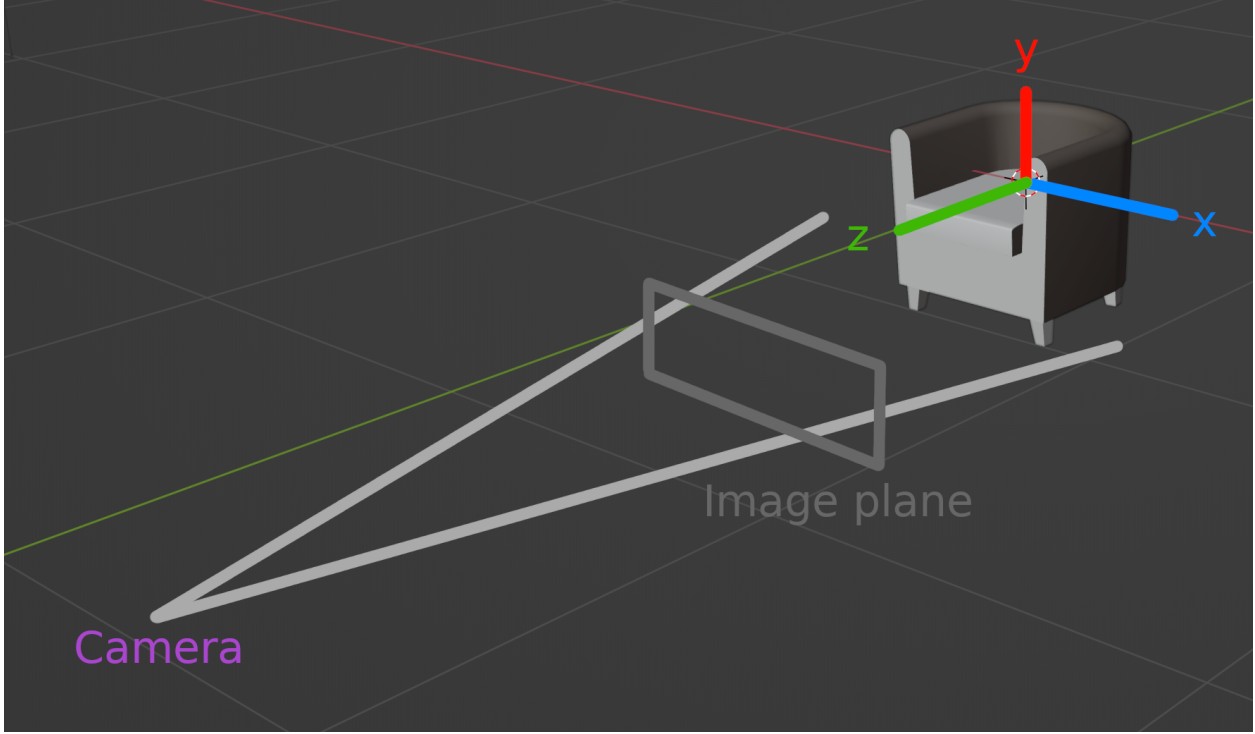

Figure 11: **Camera Setup.** This figure illustrates the camera setup in the scene, and how it relates to the captured 2D views of the object. The object pose is described by 6 parameters, three defining the object translation while the other three defining the object rotation. Pitch, yaw, and roll correspond to rotation along the x-axis, y-axis, and z-axis respectively. A perfect recognition system should be invariant under variations of all 6 pose parameters. Approximate invariance for all three translation parameters and roll can be achieved by using random cropping and rotation augmentations.

## A  Training Hyperparameters

We trained ResNet-18 (He et al., 2016), VGG-11 (with batch-normalization) (Simonyan & Zisserman, 2014) and ViT-B/16 (Dosovitskiy et al., 2020) architectures with nearly identical hyperparameters. We summarize all hyperparameters in Table 1. All our models are trained for 300 epochs with a batch size of 128 using SGD with a momentum of 0.9, a learning rate of 0.1 for ResNet-18, and a learning rate of 0.01 for VGG-11 and ViT-B/16, cosine learning rate decay, and a weight decay of 1e-4. We use gradient clipping threshold of 10 for training all our models following the training recipe for ViTs (Dosovitskiy et al., 2020). We use ImageNet-21k pretrained weights for ViT-B/16 from TIMM (Wightman, 2019) due to the difficulty of training them from scratch (Dosovitskiy et al., 2020).

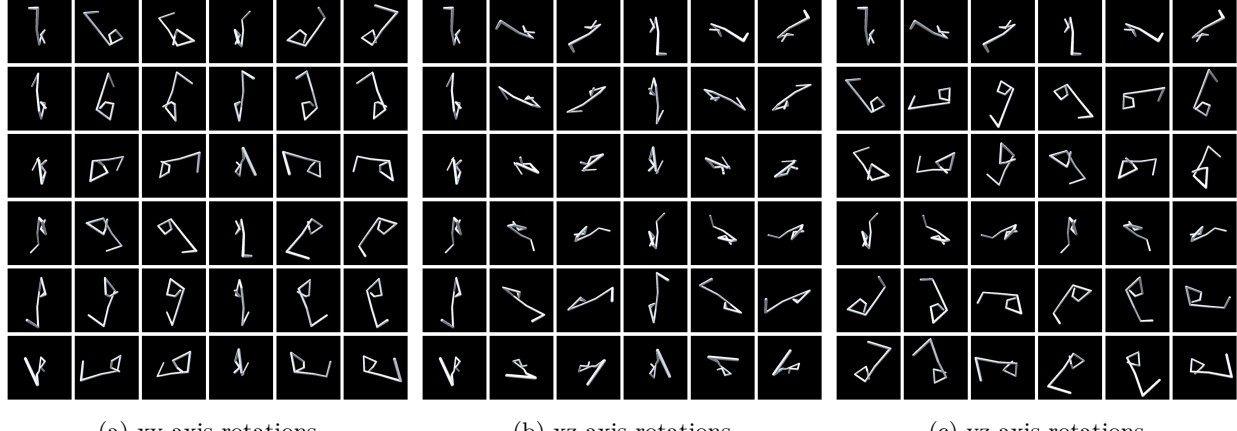

(a) xy-axis rotations      (b) xz-axis rotations      (c) yz-axis rotations

Figure 12: **Examples from the generated Paperclip Dataset.** We visualize rotations at a stride of 60 degrees for all three multi-axes rotations, where the first letter indicates rotations along the vertical axis, while the second letter indicates rotation along the horizontal axis.

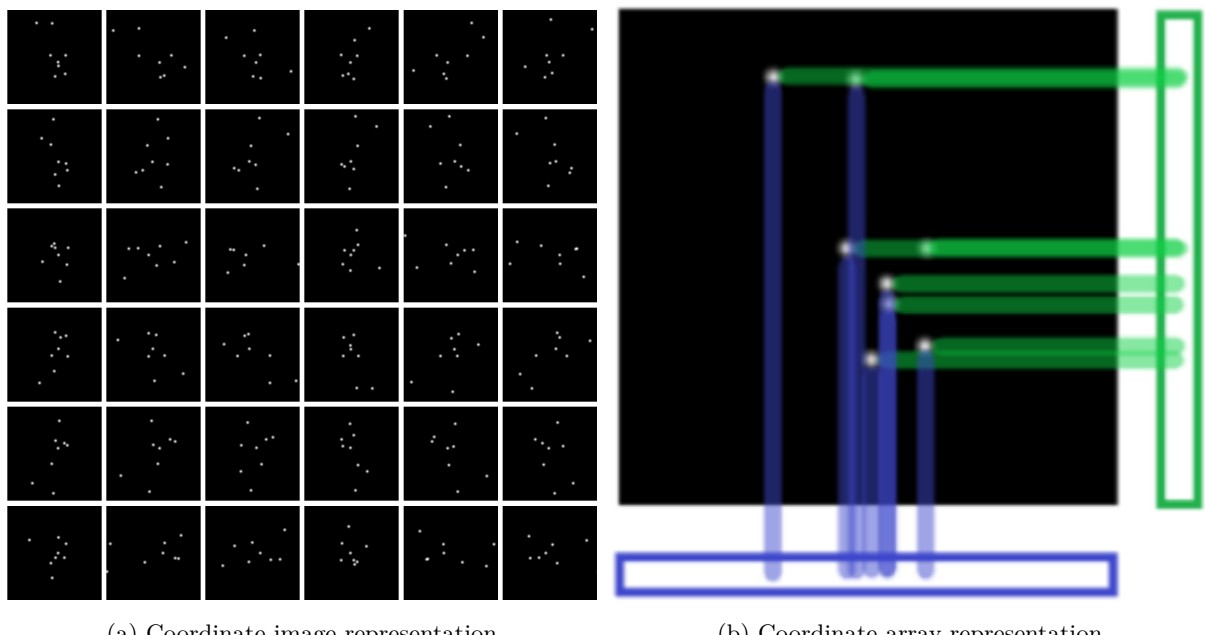

(a) Coordinate image representation      (b) Coordinate array representation

Figure 13: This figure presents an overview of the new representations used for model training. Coordinate image representation is similar to the full paperclip images, except that only the vertices of the paperclip are represented. The coordinate array representation projects the coordinates to two 1D arrays, which are concatenated together to form the final array representation.

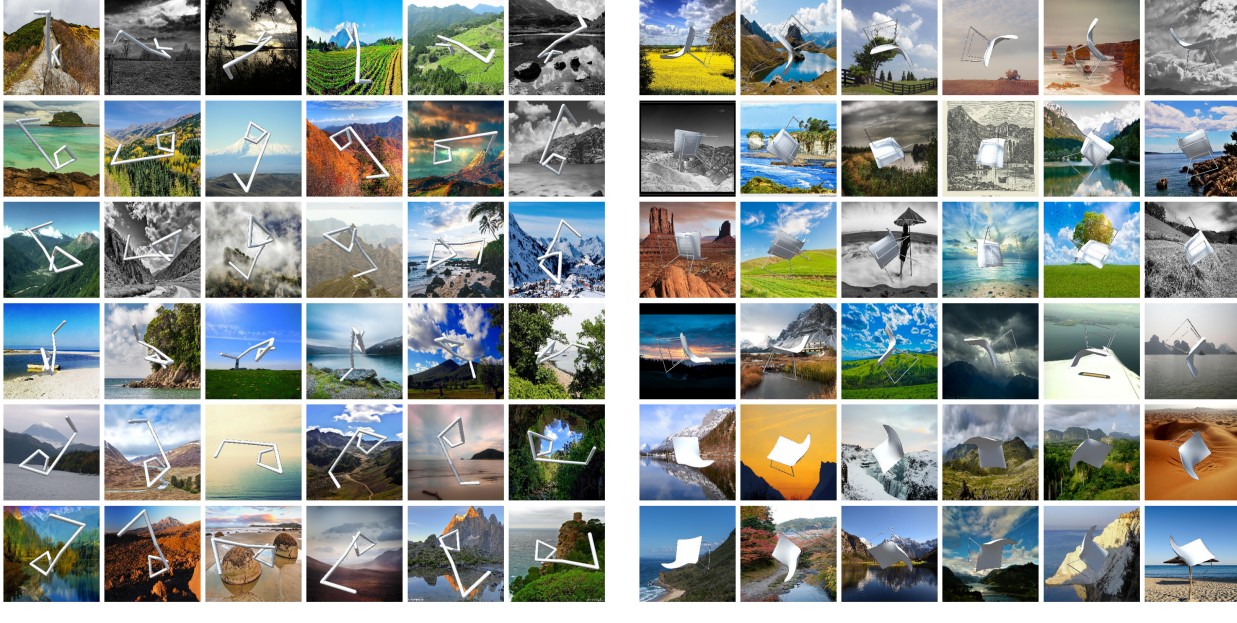

(a) Paperclips dataset

(b) Chairs dataset

Figure 14: **Datasets w/ Random Backgrounds.** We replace the black background in the original datasets with random landscape background images taken from `https://www.kaggle.com/datasets/arnaud58/landscape-pictures/` in order to highlight that the obtained generalization performance is not an artifact of the chosen black background.

