# OpenReview forum: "Investigating the Nature of 3D Generalization in Deep Neural Networks"
_TMLR — Rejected by TMLR_

### Review · Reviewer_NvDj · 2023-11-06

**Summary Of Contributions:**

This work analyzes the generalization ability of deep learning systems to 3D rotations of objects. In particular the authors generate training sets of 3D objects with different 3D transformations applied to them and set up the training objective for the machine learning system to be recognition of the specific object instance. Given these premises they subsample the training sets to show to the models only a subset of all the possible 3D transformations and study how well the models generalize to the missing ones. Unsurprisingly the models do not unless the unseen transformations are relatively close to the one seen during training. Other insights are that when training with many more objects the models can generalize slightly better, hinting at some re-use of knowledge among different classes. Experiments are performed on MLP, CNN and ViT providing similar results.

**Audience:**

No

**Broader Impact Concerns:**

No concern

**Claims And Evidence:**

Yes

**Requested Changes:**

In my opinion this paper lacks in content and novelty for being accepted in the current format. The change requested to improve it will basically account to extend it significantly to make a different work out of it. In particular my suggestion to the author is to either focus on a more broad study on the generalization limitation of Ml models to 3D transformation considering more realistic training sets and use cases (e.g. 6-D pose estimation?), to expand the analysis to also incorporate the influence of the training objective in the development of 3D generation capabilities and, finally, to propose also some mitigation measures to improve the generalization.

**Strengths And Weaknesses:**

## Strengths

+ The details of the study are very clear and the insights extracted have been nicely presented.


## Weaknesses

a. **Relevance**: I’m not sure what this work adds to concepts which are already well known by anybody working with ML models in the field of computer vision. The results of the analysis of the paper is that models do not extrapolate to unseen 3D transformations and can interpolate only up to a certain point. The other insight is that more data helps (either in terms of more transformations seen during training or more classes used to create the training set). To me these are not novel concepts.

b. **Limited analysis on the problem formulation**: All the experiments in the work uses an instance classification objective to train their model. It’s unclear how much this contributes to pushing the model to “ignore” the 3D geometry of the object. For example can a model trained to regress the pose of objects generalize better to unseen poses? I would have expected a journal paper to run way more detailed experiments covering also these aspects.

c. **Very toy experiments**: I consider the settings used in this work for the experiment to be a bit unrealistic for what concerns not having access at all to a whole category of 3D transformations. The settings followed in the previous work by [Cooper 2021 ] are more relevant for investigating the generalization performance of neural networks wrt 3D transformations. In their settings the authors consider having access to full 3D transformations for a subset of models, while for others they did not. This is a setting closer to practical use cases than the one considered in the work.

d. **No mitigation strategy proposed**: The paper does not propose any mitigation strategy for the problem highlighted, therefore diminishing the contribution of the work.

---

> ### Author Response · Authors · 2024-01-05
> **Response to reviewer NvDj**
>
> First of all, we would like to thank the reviewer for their precious time in reviewing our manuscript and for their constructive feedback. We attempt to clarify any confusion and answer the questions asked by the reviewer below:
>
> > This paper lacks in content and novelty for being accepted in the current format
>
> **Response:** Our work is aimed at understanding the nature of 3D generalization given a small number of training views, following the experiments performed on humans and monkeys primarily in psychology and psychophysics. The reviewer questions the novelty of our results.
>
> Our results show rapid generalization from a small number of views in deep learning models, not accounted for by any widely used formal models of 3D generalization, as well as enhanced generalization with the number of object classes, independent of the architecture. These phenomena have not been measured before and are unexpected and unexplained by prior formal models of 3D recognition and generalization in computer vision.
>
> These results are different from OOD generalization evaluated in other works, which measures the generalization of existing classes in other domains. Furthermore, the improvement in generalization observed in prior works is considered with an increasing number of instances per class, while our results show improvement with an increasing number of classes while keeping the number of instances per class intact.
>
> Therefore, we believe that our results are both novel and interesting. Even if the reviewer believes our work to be just a confirmation of existing concepts in machine learning, we consider TMLR to be the right venue as it focuses on correctness over novelty: https://jmlr.org/tmlr/acceptance-criteria.html
>
> > Limited analysis on the problem formulation: All the experiments in the work uses an instance classification objective to train their model. It’s unclear how much this contributes to pushing the model to “ignore” the 3D geometry of the object. For example can a model trained to regress the pose of objects generalize better to unseen poses? I would have expected a journal paper to run way more detailed experiments covering also these aspects.
>
> **Response:** We agree with the reviewer that other formulations aside from classification might provide a different picture. However, our experiments are primarily inspired by the experiments in psychology and psychophysics performed on monkeys as well as humans, which were formulated in the form of classification. Therefore, our experiments are well-grounded from prior studies.
>
> > Very toy experiments: I consider the settings used in this work for the experiment to be a bit unrealistic for what concerns not having access at all to a whole category of 3D transformations. The settings followed in the previous work by [Cooper 2021 ] are more relevant for investigating the generalization performance of neural networks wrt 3D transformations. In their settings the authors consider having access to full 3D transformations for a subset of models, while for others they did not. This is a setting closer to practical use cases than the one considered in the work.
>
> **Response:** This criticism is difficult to understand, given that Cooper et al. (2021) uses models and views similar to ours. The assumption that full 3D transformations are available for a subset of models is, if anything, less realistic, given that current object recognition models are usually trained only on views randomly chosen from the environment without knowledge of, or control over, the 3D transformations. Cooper et al. (2021) uses a similar setting to ours, but focuses on analyzing neural mechanisms and tests different hypotheses. We believe the results are complementary.
>
> > No mitigation strategy proposed: The paper does not propose any mitigation strategy for the problem highlighted, therefore diminishing the contribution of the work.
>
> **Response:** The observations have two clear, novel, immediate implications for work in object recognition using deep learning: (1) error rates from experiments on datasets with small numbers of classes cannot be directly compared to experiments on datasets with large numbers of classes, and (2) to improve 3D generalization, a simple and effective strategy is to increase the number of object classes the system is trained on. We believe that these are two important, practical insights.

---

### Review · Reviewer_hXo9 · 2023-11-09

**Summary Of Contributions:**

This paper presents a study of neural nets' generalization to new rotation angles of a training object. The authors create a synthetic dataset of paperclips and chairs viewed at angles spanning certain ranges, and then they test the model's capability on seen and unseen angles, across axis sweeps that were partially or entirely unseen. The study is framed as figuring out which of three hypotheses is true, and by ruling out two of the hypotheses the authors claim the third is basically true.

**Audience:**

Yes

**Claims And Evidence:**

No

**Requested Changes:**

It would be great if the hypotheses were revised or replaced with better ones.

It would be great if all of the clarity issues I mentioned (and ones I did not mention) were resolved by rewriting those parts of the paper.

It would be better if the figures made sense on their own, without needing to refer to the main text. For example, the second last row in Figure 2 is not well explained in the caption.

Figure 2a-c seems to have an error. The caption says "the gray line indicates the expected performance from a purely view-based model", and this gray line is hovering near zero across all degrees, instead of shooting toward 100% accuracy at the training views.

**Strengths And Weaknesses:**

I appreciate the thoroughness of the experiments. The suite of experiments presented here must have been very expensive to compute. The plots are beautiful, and for the most part, clear. The results make sense. While not particularly creative or interesting, the experiments confirm what I think many people believe already about neural nets' generalization to new viewpoints.

One overall weakness of this work is that there is nothing surprising or particularly interesting about the results. The results presented here basically consist of a clean exploration of the performance landscape, which I think most people working in the area are already familiar with.

Another problem is the three hypotheses, and the fact that one is picked as the winner. These hypotheses, pulled from cognitive psychology literature, connect poorly with the phenomenon at hand -- as demonstrated in the experiments. The third hypothesis, "linear combination of views", might in some sense be the least bad of the three, but it is never clearly defined in a way that can be proven true or false. The paper essentially concludes by saying that the first two are definitely wrong, so maybe the third is true. This is not good reasoning.

A subtler issue here is clarity. For example the word "model" is used ambiguously in a variety of contexts, sometimes meaning hypothesis/explanation, sometimes neural network, and sometimes 3D object. The paper mentions "2.5D" in parentheses as an alternate description of "linear combination of views", but to me these are very different. The paper claims difference from Cooper et al., by saying that Cooper evaluated on out-of-distribution poses, whereas this work evaluates generalization to unseen poses -- to me these are the same thing. The paper mentions creating 50M training samples but then says they subsample from here, making the 50M a waste of resources. Numerous times the paper talks about "classes" (of which there are 10k), but it is unclear what the classes are, and probably they are not categories, since all of the experiments happen with either paperclips or chairs. Lastly, it is never made clear what the training objective is for the neural nets. There are more issues, but these stand out to me the most.

---

> ### Author Response · Authors · 2024-01-05
> **Response to reviewer hXo9**
>
> First of all, we would like to thank the reviewer for their precious time in reviewing our manuscript and for their constructive feedback. We attempt to clarify any confusion and answer the questions asked by the reviewer below:
>
> > There is nothing surprising or particularly interesting about the results
>
> **Response:** The behavior we observed has not been previously reported in the literature (and the reviewer has pointed to no references). Whether it is “surprising” is, of course, a matter of opinion. A reasonable hypothesis is that with an increasing number of classes, the potential of confusion between novel views and other classes increases, so that generalization to novel views decreases. An alternative hypothesis is that 3D recognition occurs through a form of “recognition by parts” so that additional classes contribute to improved parts models, and hence improved generalization. Which hypothesis describes deep learning models better is an empirical question, which our paper resolves experimentally; it cannot be resolved by opinion.
>
> We note that our results are different from OOD generalization evaluated in other works. Furthermore, the improvement in generalization observed in prior works is considered with an increasing number of instances per class, while our results show improvement with an increasing number of classes while keeping the number of instances per class intact.
>
> We also note that TMLR would be the proper venue to publish confirmatory results even if they are “unsurprising”: https://jmlr.org/tmlr/acceptance-criteria.html
>
> > Another problem is the three hypotheses, and the fact that one is picked as the winner. These hypotheses, pulled from cognitive psychology literature, connect poorly with the phenomenon at hand -- as demonstrated in the experiments. The third hypothesis, "linear combination of views", might in some sense be the least bad of the three, but it is never clearly defined in a way that can be proven true or false. The paper essentially concludes by saying that the first two are definitely wrong, so maybe the third is true. This is not good reasoning.
>
> **Response:** The three models/hypotheses we compare against are the three major formal models of 3D recognition used in psychology and throughout the history of computer vision. The fact that deep learning-based systems behave differently from all of them is an important result. The next step is to see whether we can develop formal models that connect 3D object structure with recognition performance in a way that mirrors deep learning network behavior.
>
> We don’t conclude that “maybe the third is true”. None of the three formal models account for the behaviors we observe; none of them are “true”. Linear combination of views is merely the most similar to what we observe, but it fails to account for the lack of extrapolation and the effects of increasing number of classes.
>
> > 50M training samples but then says they subsample from here, making the 50M a waste of resources
>
> **Response:** We subsample only for training. All the examples are used for evaluation (dependent on the number of classes considered).
>
> > Figure 2a-c seems to have an error. The caption says "the gray line indicates the expected performance from a purely view-based model", and this gray line is hovering near zero across all degrees, instead of shooting toward 100% accuracy at the training views.
>
> **Response:** Thanks for highlighting this confusion. One gray line is added per view. Therefore, we see the remaining gray lines to be at zero, while one of them is at 100%.

---

> > ### Comment · Reviewer_hXo9 · 2024-01-23
> >
> > I said: "The paper essentially concludes by saying that the first two are definitely wrong, so maybe the third is true. "
> >
> > The response said: "We don’t conclude that “maybe the third is true”. None of the three formal models account for the behaviors we observe; none of them are “true”. Linear combination of views is merely the most similar to what we observe, but it fails to account for the lack of extrapolation and the effects of increasing number of classes."
> >
> > This would be OK, but actually the paper says:  "These results disprove both full 3D recognition and pure 2D matching as possible classes of behavior employed by deep models, **leaving only the linear combination of views as the most likely class of behavior exhibited by deep models**." To me, this sounds like my original summary. If the authors agree that the third hypothesis is not likely to be true, perhaps this claim can be revised.
> >
> > About the gray lines in the figure: OK, looking closely, I can see that the plots have an increasing number of gray lines, starting with zero in the first plot. I am not really sure why this is the case -- the caption just says "the gray line" in the same sentence as "the red line", so I expect one gray line per plot.

---

> > > ### Author Response · Authors · 2024-01-26
> > > **Reply to reviewer hXo9**
> > >
> > > We are thankful to the reviewer for further clarifying the confusion. We understand now why the reviewer assumed us to be selecting one hypothesis out of the three. This is merely a confusion due to the writing style. In section 4.1, we mentioned (as the reviewer highlighted): *“These results disprove both full 3D recognition and pure 2D matching as possible classes of behavior employed by deep models, leaving only the linear combination of views as the most likely class of behavior exhibited by deep models.”.*
> > >
> > > This wasn’t our approval, but rather, we were trying to indicate the fact that we have disproven two hypotheses, leaving only the last one to be evaluated. Therefore, in section 4.2, we mentioned: *“Results in Section 4.1 already ruled out the possibility of pure 2D matching or full 3D recognition, making the linear combination of views the most likely class of behavior employed by deep models for recognition. However, for objects such as paperclips, where self-occlusion is not an issue, a linear combination of views should be able to extrapolate i.e. generalize within the axis of rotation. Since this is not the case, this indicates that the model is operating somewhere close to the linear combination of views, but is still distinct in some aspects.”.*
> > >
> > > Since there was a risk of confusion as the reviewer highlighted, we have updated the text in section 4.1 to avoid this confusion (marked in RED). Here is the updated version for section 4.1: **“These results disprove both full 3D recognition and pure 2D matching as possible classes of behavior employed by deep models, leaving only the linear combination of views as a possible class of behavior exhibited by deep models, which we will investigate further.”**

---

### Review · Reviewer_MyGP · 2023-12-09

**Summary Of Contributions:**

The paper performs an evaluation of how well image processing neural networks generalize to 3D viewpoints. The paper further analyses the mechanism by which this generalization is performed by comparing with baselines on pure 2D matching as well as matching based on a linear combination of the views. The paper tests a variety of different architectures.

The paper shows that generalization of vision models is not explained by building global 3D models, and neither by only matching images directly. The paper rather shows that the behavior is better explained by a baseline that performs linear combination of input images (the exact nature of this baseline, however, is unclear!).

**Audience:**

Yes

**Broader Impact Concerns:**

N/A.

**Claims And Evidence:**

Yes

**Requested Changes:**

- The authors have to clarify - ideally mathematically, with a set of equations (which I assume should be very simple) - what they mean by the "linear combination of input views" baseline.
- I would advise - but not request - that the authors put in the extra effort of putting up a github repository that enables one to quickly run the same evaluations with *any* model. I.e., if these experiments were implemented in pytorch (which I assume they were), then the authors should offer a codebase that enables other research groups to plug in any pytorch model, press "run", and produce the same set of results that the authors produce in this paper. This would drastically increase the impact of the present work, and make the fruits of the authors' labor accessible to other research groups.
- The authors need to include a discussion of neural tangent kernels and the theory of generalization in CNNs.

**Strengths And Weaknesses:**

## Strengths
- The paper does a good job at stating hypotheses, assumptions, and evaluation protocol.
- The evaluation protocol is thorough.
- Results are presented in a visually pleasing and easy-to-parse manner.
- The presented experiments could serve as a good benchmark of out-of-distribution generalization for future work.

## Weaknesses.
- I did not understand what the authors mean by "linear combination of input views", and how the authors would build a classifier that would operate in this manner. As far as I can tell, the authors fail to explain that mechanism. I went back to the paper by Ullman and Basri, but it seems that that paper assumes that one knows the *coordinates* of edges / has correspondences of images across views. The present paper does not detail whether they literally linearly interpolate the views, or whether they linearly interpolate the coordinates of paperclip correspondences.
- The authors fail to discuss any of the theory behind generalization of neural networks. References largely stem from the cognitive psychology and human perception line of research. However, for Convolutional Neural Networks (CNNs), the community has worked out a principled framework for understanding generalization, which is that of neural tangent kernels (NTKs). Intuitively, the neural network acts as a kernel machine: given a test image, it will compute a similarity to training images, and output the test-time log-likelihoods as a linear combination of the training labels. In the case of CNNs, we know that this kernel machine is one that compares image patches, with a locality bias. See for instance Tachella, "The Neural Tangent Link Between CNN Denoisers and Non-Local Filters", for a discussion. This discussion is critical, as it provides a principled perspective on the results presented in the paper - indeed, the CNN will simply match patches in the test image to patches in the training set. However, this is in an idealized context, and further, a similar theory (to the best of my knowledge) does not exist for VITs.

---

> ### Author Response · Authors · 2024-01-05
> **Response to reviewer MyGP**
>
> First of all, we would like to thank the reviewer for their precious time in reviewing our manuscript and for their constructive feedback. We attempt to clarify any confusion and answer the questions asked by the reviewer below:
>
> > The authors have to clarify - ideally mathematically, with a set of equations (which I assume should be very simple) - what they mean by the "linear combination of input views" baseline.
>
> **Response:** Thank you for raising this. Linear combination of views assumes that the object has been correctly segmented, and uses coordinates of the edge-map of the input as the reviewer understood to be the case. A given image P is assumed to match a model M^{i} represented by views M^{i}={m_{1}^{i}, m_{2}^{i}, .., m_{k}^{i}} if for any constraints \alpha_i, P = \sum_{j=1}^{k} \alpha_{i} m_{j}^{i}. We have updated the description in the paper to make this more clear.
>
> We updated the description of the linear combination of views model in the draft to make this clear (section 2.2). The changes are marked in RED for better visibility.
>
> > The authors fail to discuss any of the theory behind generalization of neural networks
>
> **Response:** We understand and agree with the reviewer’s concern. Regardless of the details of the generalization, neural networks with image inputs generalize in image space, which is rather different from coordinate space. The kind of generalization we see is not easily explainable merely by image similarity. This is primarily why we looked at coordinate representations in Fig. 5.
>
> > The authors need to include a discussion of neural tangent kernels and the theory of generalization in CNNs.
>
> **Response:** NTKs represent an interesting way of representing generalization in neural networks. But for any form of generalization, there is some NTK that describes it, so it is unclear to us what an analysis in terms of NTKs would add. It is possible that in future experiments, NTKs might become a useful way of describing/analyzing results.
>
> > I would advise - but not request - that the authors put in the extra effort of putting up a github repository that enables one to quickly run the same evaluations with any model. I.e., if these experiments were implemented in pytorch (which I assume they were), then the authors should offer a codebase that enables other research groups to plug in any pytorch model, press "run", and produce the same set of results that the authors produce in this paper. This would drastically increase the impact of the present work, and make the fruits of the authors' labor accessible to other research groups.
>
> **Response:** Thank you for the valuable suggestion. We do have a GitHub codebase that allows plugging in any model of choice for evaluation. We will add a link to our repository in the paper after the end of the double-blind review period.

---

### Decision · Action_Editor_vLuw · 2024-01-28

**Recommendation:** Reject

**Comment:**

The AE agrees with the final comments from the reviewers, who all recommended rejection due to the insufficient validation of the claims on analyzing deep networks' 3D (novel view) generalization and the limited relevance to the TMLR community.

Reviewer NvDj pointed out that the main claims of the submission on analyzing neural networks' 3D generalization should be validated through a formal study of the phenomena; however, the current evaluation on limited cases, especially without results on in-the-wild natural images, is insufficient to support the broad claims.

Reviewer MyGP further pointed out that the task designed for neural networks is not a commonly studied problem in the field. Therefore, the input-output behavior of the network is not a model of a typical 3D vision model. Therefore, the results from the submission cannot support the claims from the submission on analyzing the 3D generalization of neural networks.  The reviewer also pointed out issues on not discussing the related work on NTK. They said, "My point was that the generalization behavior of the neural network in question should be seen through the angle of the NN basically "learning" a kernel function over the input coordinate space. If one understands that kernel function, then one understands the generalization behavior of the network. It seems that that ought to be at least worth a discussion...?"  The AE agrees with the comment.

Considering all these factors, the AE recommends a rejection.

**Audience:**

The topic itself would be interesting to some of the TMLR audience.  However, the limited experiments and conclusions in the current submission have prevented it from appealing to a larger community who may otherwise be interested in the submission.

**Claims And Evidence:**

The submission aims to analyze deep networks' 3D (novel view) generalization.  Unfortunately, all three reviewers felt that the claims on characterizing the ability of common deep learning architectures to generalize to novel views were not well supported, primarily due to the limited stimuli used in the experiments.  Reviewers were also concerned about the missing discussion of related work, such as NTKs.